# Examination of factors that impact mask or respirator purchase and usage during the COVID-19 pandemic

Nicole Bertges[1], Sachiyo Shearman[2], Satomi Imai[1], Jo Anne G. Balanay[3], Sinan Sousan[1,4,5]*

1 Department of Public Health, Brody School of Medicine, East Carolina University, Greenville, North Carolina, United States of America, 2 School of Communication, East Carolina University, Greenville, North Carolina, United States of America, 3 Department of Health Education and Promotion, Environmental Health Sciences Program, College of Health and Human Performance, East Carolina University, Greenville, North Carolina, United States of America, 4 North Carolina Agromedicine Institute, Greenville, North Carolina, United States of America, 5 Center for Human Health and the Environment, NC State University, Raleigh, North Carolina, United States of America

* sousans18@ecu.edu

**Data Availability Statement:** The data supporting the findings of this study are available from the

## Abstract

Understanding the motivations and barriers populations face in respiratory protection during a pandemic is key to effective primary prevention. The goal of this study was to identify the impact of various motivational factors on individual choice of masks or respirators during the COVID-19 pandemic. The survey study included a sample of 619 participants between the ages of 18 to 75 years old and English speaking, who were asked about factors that affected their choice of masks or respirators between the months of April 2020 and May 2021. Participants showed a positive association between choice of respirator and being male, perceived comfort and trust in respirators, importance of personalization, and trust in information from family or friends. In addition, positive associations were found between income, education, and the amount of trust in CDC, WHO, and local/state health departments. Results indicate the need for increased education on attributes of each mask or respirator, review of respirator fit, research into specific areas of discomfort, and outreach efforts to vulnerable populations.

## Introduction

In January 2020, COVID-19 disease, emerging most probably from in Wuhan, China, affected the world within months. By March 2020, the World Health Organization (WHO) declared COVID-19 a pandemic and global emergency [1]. Masking is a universally effective prevention method that helps prevent the transmission of respiratory diseases such as COVID-19 [2–6]. The need for masking created an unprecedented mask shortage, and in an effort to aid in supply shortages in the United States, the Centers for Disease Control and Prevention (CDC) developed guidelines that included the reuse of respirators and recommended alternate types of protective wear, such as cloth masks [7]. Public health communication during the pandemic

Open Science Framework (OSF) at the following link: https://osf.io/c8je9/.

**Funding:** This research was supported by the Department of Public Health, School of Communication, and Department of Health Education and Promotion at East Carolina University; the North Carolina Agromedicine Institute; and the National Institute of Environmental Health Sciences of the National Institutes of Health under Award Number P30ES025128. The content is solely the responsibility of the authors and does not necessarily represent the official views of the National Institutes of Health.

**Competing interests:** The authors have declared that no competing interests exist.

became a challenge due to the influence of social media and political agendas. Mask or respirator use subsequently ignited a polarizing public debate concerning their benefits, personal disbeliefs, and masking behavior during the COVID-19 pandemic.

At the height of the COVID-19 pandemic, there were many options for masks or respirator choices, each offering differing protection levels and comfort. The choice of respirators, such as the N95 or the KN95, and masks, such as surgical masks or cloth masks, determines the amount of protection offered to the wearer and the individual's comfort level. Multiple studies have shown that N95 respirators provide superior protection [2–4,6,8,9]. The alternative respirator from China, the KN95, can provide protection against respiratory disease, but it is not certified by the National Institute for Occupational Safety and Health (NIOSH) [10]. While respirators offer superior protection, alternatives such as surgical and cloth masks have been shown to offer some degrees of protection based on material, wearer behavior, reuse, or fit [4–6,11].

While the quality of the product should be the deciding factor in choosing a mask respirator, the choice is often much more complex. Demographic factors such as age, race, and sex have been shown to be associated with masking behavior. In a study conducted by Haischer et al. (2020), masking adherence increased with age and was higher in females [12]. Race has also been shown to be associated with masking behavior, where a study conducted by Pecoraro et al. (2022) reported that masking was less prevalent in white populations [13]. An individual's perceptions about masks or respirators have been shown to be associated with masking behavior [14]. Cooks et al. (2022) showed that the source of information on masks or respirators and COVID-19 played a role in making decisions, and that race, income, and education influenced which sources were perceived to be more trustworthy [15].

In many individuals, comfort, such as the breathability and the amount of heat or moisture retained in the mask are cited as a key factors in a mask or respirator choice [16–20]. Cost, convenience, and availability of mask or respirator supplies have been cited as factors in mask choice [21–26]. Additionally, the risk of infection a person faced during the COVID-19 pandemic can affect their masking behaviors. Wismans et al. (2022) reported a positive association between an individual's belief of their risk of COVID-19 and masking behavior [27]. Race, education, and income are factors that could influence the amount of trust some populations have in important sources of information, such as the CDC, WHO, and their local or state health departments. Issues of digital access, the perception of mixed messages, and potential xenophobia can create a feeling of mistrust in many, especially in vulnerable populations, on critical issues, such as which masks to choose [15].

This study examines the behavioral motivations behind individual mask/respirator choices, building on the work of Chaaban et al. (2023), who compared best-selling alternative masks and respirators to the N95 respirator in terms of cost, breathability, and the amount of protection against viral particles in the event of future shortages [17]. This study aims to identify the factors that influenced participants to choose the higher protection offered through the N95 and KN95 respirator or the lower protection offered through the cloth and surgical masks. The authors identified six factors that could influence the choice of high-protection respirators or low-protection masks.

The study investigated the role of 1) demographics; 2) an individual's perceptions and attitudes on factors relating to masks and respirators; 3) an individual's perceived level of trust toward the source of information; 4) an individual's belief in the protective ability of masks or respirators against the COVID-19 virus; 5) an individual's perceived risk and protective practices (such as vaccination, handwashing, and social distancing); and 6) an individual's perceived comfort on the choice of mask or respirator type (high or low protection). In addition, the role of demographic factors such as race, income, and education on the amount of trust in

official information sources (CDC, WHO, or local and state health departments) is investigated. The goal of this research is to provide information on respirator and mask usage to the public that addresses an individual's personal motivations and can provide guidance that considers the limitations and barriers they may encounter in choosing their respiratory protection.

## Methods

### Research design

The study was a cross-sectional study design, and consisted of a survey administered through the Qualtrics Research Platform. The survey questions, written to examine factors present during the peak of the pandemic from April 2020 to May 2021, consisted of multiple-choice questions or Likert scale responses. The sample size was determined through a combination of survey sample best practices, financial considerations, and calculation of recommended sample size based on ±5% Margin of Error and the large study population size [28].

### Procedures/Measures

The data collection protocol was approved as an exempt study (UMCIRB#22–002104) by the East Carolina University Medical Center Institutional Review Board on March 22, 2023. The participants were given written informed consent before participating in the study. The consent form included a full explanation of the study's purpose, how the information gathered would be used, and their ability to terminate their participation at any time. Then, participants provided online consent by clicking "agree to participate" in the online survey. Only de-identified information was gathered from participants, and the survey responses were protected using the security settings provided by the Qualtrics Research Platform. Data collection was between March 22nd, 2023, and March 27th, 2023. An initial pretesting on a smaller population was conducted, which allowed the authors to identify possible wording bias in the survey questionnaire. The sample inclusion criteria was set as individuals above the age of 18 and English speakers. The survey asked the demographic questions such as age, sex, race/ethnicity, and annual household income. In addition, the survey asked the participants' preferences of respirator and mask usage, various factors in selecting masks and respirators, perceived comfort, perceived protection, perceived risks and protective practices, and perceived level of trusts toward the sources of information on masks and respirators. Included basic demographic questions such as age, sex, race, annual household income A convenience sample was gathered through the Qualtrics Research Platform. Individuals were previously recruited to a survey panel and sent an email/link asking them to complete the survey. Survey participants were compensated for serving on survey panels by Qualtrics Research Platform. The participants did not receive any compensation by the authors in this study.

### Participants

A total of 619 participants participated in the survey, with the selection criteria set to ensure that the participants would be balanced based on age and gender. Males made up 50.7% (n = 314), 49.9% (n = 303) were female, and two individuals identified their gender as others, but were removed from the analysis due to the small sample size. The age distribution showed that 19.5% were between 18 and 30, 45.7% were between 31 and 50, and 34.7% were between 51 and 75. The majority of current study's participants identified themselves as non-Hispanic white (69.2%). Only 3.2% of the participants had less than a high school diploma, compared to 21.5% with a high school diploma or equivalent, 6.3% associate degree, 30.5% bachelor's

degree, 16.3% master's degree, and 2.3% with a doctoral degree. Annual income of the sample population reflected that 33.4% made less than $35,000, 28.9% between $35,000 and $75,000, and 37.6% made more than $75,000.

## Statistical analysis

Statistical analysis was performed using IBM SPSS Statistics (Version 28). The analysis investigated the factors that influenced the participants' choice of mask (surgical or cloth) or respirator (N95 or KN95), as well as the amount of trust disadvantaged populations (Black/African American, Hispanic, low-income, and low-education) had in the major sources of information on masks, respirators, and the COVID-19 virus during the pandemic. The variables analyzed are shown in Table 1 and were divided into the categories of style, comfort, convenience, choices, individual perceptions, the source of trusted information on masks and respirators or COVID-19, education, income, race, vaccination status, and sex. The independent variables were assessed using a combination of dichotomous, 5-point Likert scale, and multiple-choice questions. Income was dichotomized into "Under $35,000" and "Over $35,00" based on income limits reported by the U.S. Department of Housing and Urban Development's Office of Policy Development and Research [29]. Descriptive analysis was used to identify missing or inconsistent data. Variables with inconsistent scaled responses were recoded to ensure they were consistently scaled. A correlation matrix was constructed to identify multicollinearity. Prior to combining the variables, such as the N95 and KN95 into high protection and cloth and surgical masks into low protection, both a dimensional reduction and reliability analysis was run. The variables were then entered into a binary logistic regression analysis to identify the impact each had on the choice of mask or respirator.

## Results

Table 2 shows the characteristics of the participants. Participant responses indicated that 13.0% preferred the N95, 18.6% preferred the KN95, 28.8% preferred surgical masks, and 39.5% preferred cloth masks. In the sampled population, 16.5% considered themselves having a high risk of contracting COVID-19, 57% reported having family members who were at high risk of COVID-19, and 24.7% of the participants were not vaccinated.

The results of the binary logistic regressions can be found in Fig 1. The results show that males had twice the odds of choosing respirators over masks (OR = 1.978, 95% CI [1.331, 2.939], p = 0.001). Individuals who felt that personalization was slightly important had over a two time increase in odds of choosing respirators over masks (OR = 2.619, 95% CI [1.573, 4.359] p = <0.001), but individuals who felt that convenience was very important had a decrease in odds of choosing a respirator compared to masks (OR = 0.424, 95% 136 CI [.183, 0.978], p = 0.044). Individuals who reported above-average trust in media sources of information on COVID-19 or respiratory protection had a decrease in odds of choosing a respirator compared to a mask (OR = 0.400, 95% CI [0.173, 0.926], p = 0.032). Individuals who had an average amount of trust in information on COVID-19 or face protection from family or friends had an almost two-time increase in the odds of choosing respirators over masks (OR = 1.762, 95% CI [1.062, 2.926], p = 0.028). Individuals that reported having high trust in either the N95 or KN95 respirator had an over three time increase in the odds of choosing respirators over masks (OR = 3.425, 95% CI [1.894, 6.193], p = <0.001), while individuals who reported having high trust in both the N95 and the KN95 had more than two times increase in the odds of choosing respirators over masks (OR = 2.688, 95% CI [1.685, 4.289], p = <0.001). Individuals who believed that respirators are comfortable had two times the odds of choosing a respirator over masks (OR = 1.873, 95% CI [1.176, 2.983], p = 0.008).

**Table 1. Variables analyzed.**

| Respirator use vs Mask use | | N95 Respirators and KN95 Respirators | |
| --- | --- | --- | --- |
| | | Cloth and Surgical masks | |
| **Independent Variables** | | | |
| **Demographics** | Sex* | Female | |
| | | Male | |
| | Education | Less than a four year degree | |
| | | Four year degree | |
| | Income | Under $ 35,000 annually | |
| | | Over $35,000 annually | |
| | Race/Ethnicity | Non-Hispanic White | |
| | | Non-Hispanic Black/ African American | |
| | | Other (Hispanic, Asian, Native American, Alaskan Native, Pacific Islander) | |
| **Perceptions and attitudes on factors relating to masks and respirators** ** | Moisture wicking material | | |
| | Breathability | | |
| | Convenience | | |
| | Availability | | |
| | Cost | | |
| | Choices | | |
| | Brand | | |
| | Variety of choices | | |
| | Variety of sizes | | |
| **Comfort provided by each mask or respirator** ** | Perceived comfort in N95 | | |
| | Perceived comfort in KN95 | | |
| | Perceived comfort in Surgical mask | | |
| | Perceived comfort in Cloth mask | | |
| **Perception of protection offered by each mask or respirator** ** | Perceived protection offered by N95 | | |
| | Perceived protection offered by KN95 | | |
| | Perceived protection offered by Surgical mask | | |
| | Perceived protection offered by Cloth mask | | |
| **Perceived risk and protective practices** | Risk of infection or complications due to COVID-19 | Low risk of infection or complications | |
| | | High risk of infection or complications | |
| | Vaccination status | Not fully vaccinated (no vaccination or only one of the Pfizer/Moderna vaccine) | |
| | | Fully vaccinated with or without booster | |
| **Perceived level of trust toward the source of information on masks and respirators or COVID-19** ** | Media (TV, radio, internet, social media) | | |
| | Family and friends | | |
| | Job | | |
| | Universities or schools | | |
| | Medical Professionals | | |
| | Personal research | | |
| | Centers for Disease Control | | |
| | World Health Organization | | |
| | Local or state health departments | | |

*Respondents who reported other as sex were removed due to small number in population.

** Measured using Likert Scale.

**Table 2. Sample population characteristics.**

| | | |
|---|---|---|
| **Gender** | Male | n = 314 (50.7%) |
| | Female | n = 303 (49.9%) |
| **Age** | 18–30 years old | n = 121 (19.5%) |
| | 31–50 years old | n = 283 (45.7%) |
| | 51–75 years old | n = 215 (34.7%) |
| **Race** | Non-Hispanic White | n = 427 (69.2%) |
| | Non-Hispanic Black/African American | n = 93 (15.1%) |
| | Hispanic | n = 60 (9.7%) |
| | Non-Hispanic Other | n = 37 (6.0%) |
| **Education** | Less than a four-yr. degree | n = 20 (3.2%) |
| | High School Degree or equivalent (e.g., GED) | n = 133 (21.5%) |
| | Some college, no degree | n = 123 (19.9%) |
| | Associate degree (e.g., AA, AS) | n = 39 (6.3%) |
| | Bachelor's degree | n = 189 (30.5%) |
| | Master's degree | n = 101 (16.3%) |
| | Doctoral degree | n = 14 (2.3%) |
| **Marital Status** | Not married | n = 279 (45.1%) |
| | Married | n = 340 (54.9%) |
| **Employment** | Not employed | n = 196 (31.7%) |
| | Employed | n = 423 (68.3%) |
| **Annual Income** | $0-$35,000 | n = 207 (33.4%) |
| | $35,000- $75,000 | n = 179 (28.9%) |
| | 0ver $75,000 | n = 233 (37.6%) |
| **Vaccination Status** | Not vaccinated | n = 153 (24.7%) |
| | 1 dose of Johnson & Johnson | n = 62 (10%) |
| | 2 doses of Pfizer of Moderna vaccine | n = 147 (23.7%) |
| | Fully vaccinated with 1 booster | n = 93 (15.0%) |
| | Fully vaccinated with two boosters | n = 97 (15.7%) |
| | 1 dose of Pfizer or Moderna vaccine | n = 67 (10.8%) |
| **High Risk of infection or complications due to COVID-19** | Not high risk | n = 517 (83.5%) |
| | High risk | n = 102 (16.5%) |
| **Type of mask Preferred** | N95 respirator | n = 80 (12.9%) |
| | KN95 respirator | n = 115 (18.7%) |
| | Surgical mask | n = 178 (28.8%) |
| | Cloth mask | n = 244 (39.6%) |

An analysis of the association of education, income, and race to the amount of trust in prominent sources of information on masking or COVID-19 during the pandemic was then carried out and is shown in Fig 2. Individuals with a four-year degree or higher had more than two times increase in the odds of trusting the information from the CDC (OR = 2.273, 95% CI [1.647–3.138], p = <0.001), the WHO (OR = 2.464, 95% CI [1.782–3.407], p = <0.001), and their local or state health department (OR = 2.046, 95% CI [1.481–2.827], p = <0.001) compared to individuals without a four-year degree. Marginalized populations have been previously reported to have lower trust in medical institutions and government agencies [30]. Therefore, the data was analyzed to see if there was a significant difference in the amount of trust in the information from the CDC, WHO, and local/state health departments in participants who identified as Black/African American or Hispanic. The results show that there was no statistically significant association found in this sample between race/ethnicity and the

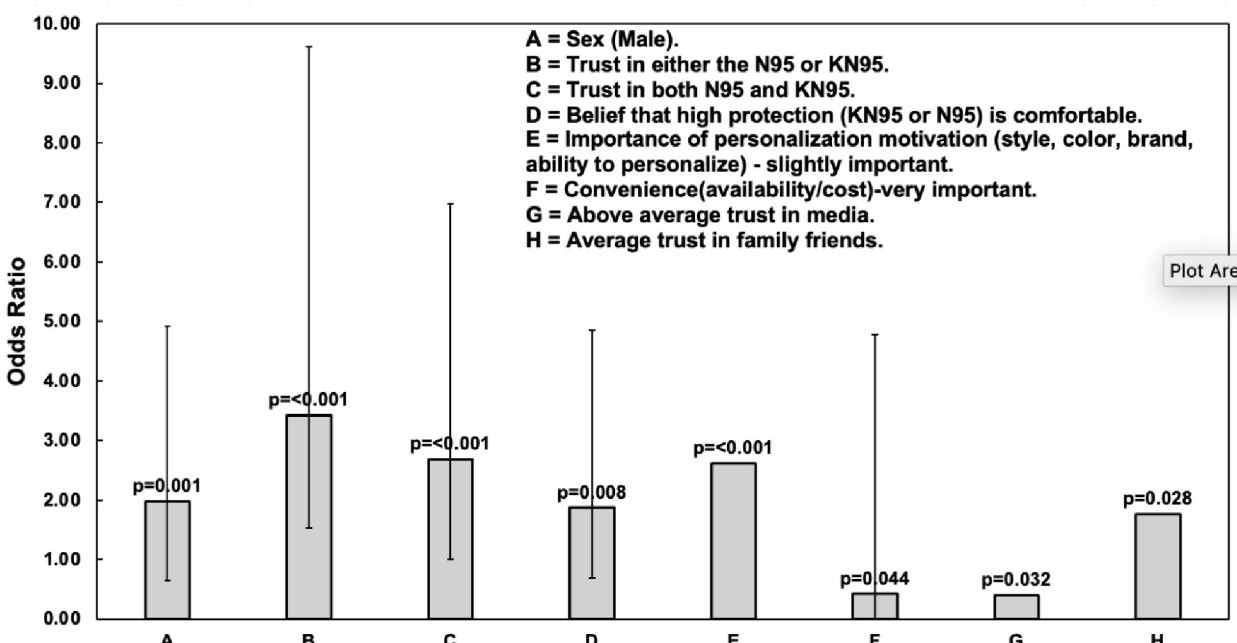

**Fig 1. Effect of individual factors on the choice of a respirator (KN95 or N95).** The error bars represent the 95% confidence intervals.

amount of trust in the CDC, WHO, or the local/state health department. However, when the data was analyzed to see if income would impact the amount of trust participants had in the CDC, WHO, and the local/state health department, the authors found that individuals who made over $35,000 were more likely to trust the CDC (Adjusted OR 1.794, 95% CI [1.277–2.520], p = <0.001), WHO (Adjusted OR 1.610, 95% CI [1.147–2.260], p = 0.006), and local/state health department (Adjusted OR 1.802, 95% CI [1.275–2.548], p = <0.001) than individuals making less than $35,000 a year.

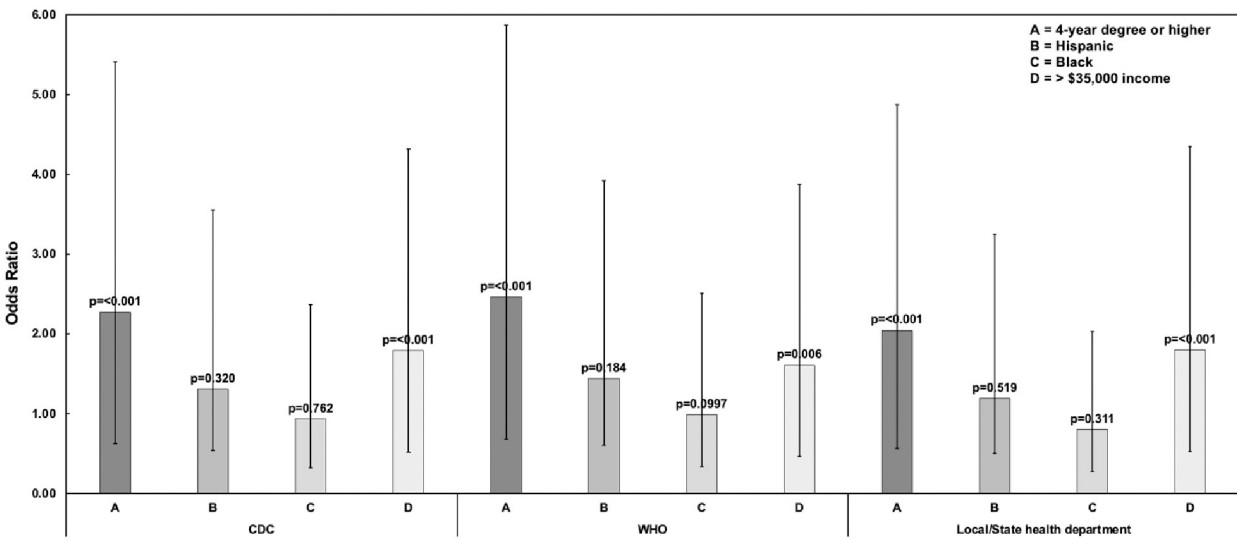

**Fig 2. Stratified analysis on effect of demographic factors on trust in sources of information on COVID-19 or prevention.** The error bars represent the 95% confidence intervals.

## Discussion

The goal of this study was to identify factors leading to the choice of respirators, KN95 or N95, and masks (cloth or surgical). The authors' aim was to provide information that could be used to develop guidance to the public in choosing the highest respiratory protection possible. Although previous studies already established that respirators, such as the N95, provide superior protection, the current study examined several factors impacting the choice of distinct types of masks and respirators. Prevention measures, such as vaccination and the interruption of modes of transmission, are vital in outbreak management. Efforts to increase community awareness of the benefits of wearing a respirator would aid in slowing the transmission of viruses and potentially minimize the number of lives lost in future pandemics. This study identified associations between the motivations of comfort, perceived protection, and trust in information sources with the individuals' choice of mask or respirator.

The study revealed that males had increased odds of choosing the higher protection provided by respirators. These results are interesting because females have been previously associated with a higher frequency of mask-wearing behavior [33]. The increased odds in males of choosing a respirator could be related to the fit of the respirators. Previous studies noted that respirators fit the faces of males better than females, due to manufacturing specifications based on a male face size or shape, which could affect comfort [31–33].

Trust in the mask or respirator's protective ability is the most crucial factor in choosing facial protection and was found to be positively associated with the choice of respirator over masks. These results could indicate that the participants choosing cloth or surgical masks are not fully aware of the amount of protection offered by each type of mask or respirator. Additionally, a positive association was found between comfort in the N95 or KN95 respirator and odds of choosing the respirator. Previous studies have associated headaches, itching, increased facial temperatures, and humidity with the 195 N95 and could be contributing to some choosing looser fitting masks over respirators [34]. The results showing that comfort is a significant motivational factor is an important finding, as it begins to address the questions of comfort posed by Chaaban et al. [17]. Further investigations into the specific areas of discomfort, role of fit, facial accessories, and environmental conditions of various demographic groups are needed to understand the motivation for comfort. Identifying specific areas of discomfort could aid in creating communications that could address this barrier and attempt to suggest ways to mitigate the discomfort without sacrificing the protection offered by respirators.

In groups with above-average trust in media sources of information on COVID-19 or face protection, there was a decrease in odds of choosing a respirator compared to a mask. This is interesting as Cooks et al. [15] associated the decrease in trust with doubts related to the efficacy of masking in general and the age of the individual. The authors' conclusion agrees with the previous findings that further research is needed on the association between masking behavior and trust in media, as the challenge of using the media will be overcoming the doubt created by the political polarizations during the COVID-19 pandemic, and mistrust in masking to address the knowledge gap in facial protection options. Interestingly, this study found a lack of significance in the association between the belief that the participant was at risk of COVID-19 and the amount of protection chosen. These results contrast with those reported by Huaman et al. [35], who reported that individuals who believed they were more likely to be at risk were positively associated with the choice of wearing an N95.

One of the most interesting findings in this study was the level of trust vulnerable population felt towards sources of information during a pandemic, such as the CDC, WHO, and their local or state health departments. The results showed no statically significant association between those identified as Black African American or Hispanic and the amount of trust in

these agencies, but there was an association with education and income. Individuals of lower income and education were less likely to trust the CDC, WHO, or local and state health departments. Previous studies into the association of education and income with trust in medical agencies have revealed mixed results [36,37]. The lack of trust in these agencies reported in this study could be due to the perception that mixed messages, due to changing guidelines, were being sent about COVID-19 and masking guidance [38]. In addition, the messages could have been communicated using scientific language that was difficult for individuals with lower education to understand. These results, and the gap in knowledge to compare it to, indicate a need to fully understand the role of income and education in the amount of trust placed in agencies, such as the CDC, WHO, and in the government during the time of a national emergency. The development of communication on masking guidance and other messages on preventive measures must be based on the ability of the target population to understand it and the likelihood of its acceptance.

## Limitations

The limitations of this study were the sample size and possible introduction of bias. The study used a convenience sample consisting of 619 participants, which limits the generalizability of the current study. When the population was identified, a filter was used to ensure that the age and sex characteristics were balanced. During this part of the study, race was not identified as a selection criterion, which might have resulted in the sample size in minority groups being too small to show statistical significance. In addition to this, the questionnaire did not include questions related to specific brands or questions dealing with the fit of the mask or respirator. As fit can affect comfort, this information could have helped to identify additional factors that led to discomfort with the mask or respirator. Future work is needed to investigate how these factors affect discomfort with masks or respirators. Brand identification could have helped to explain if the role was based on the perception of quality or merely loyalty to a specific brand name. By asking participants to identify motivations, feelings, and factors that affected their choice of mask or respirator during the peak of the pandemic, recall bias and social desirability bias could have been introduced into the study.

## Conclusion

Understanding why masks that offer lower protection are chosen is important in the development of effective exposure prevention measures during a pandemic. The authors illustrated that factors such as sex, comfort, trust, and perceived effectiveness of the mask or respirator can significantly impact the choice of high protection offered by respirators or the lower protection of masks. The knowledge gained will aid in creating targeted communications and outreach aimed at building trust in vulnerable populations, to increase the effectiveness of exposure control measures, such as the choice of higher protection offered by respirators.

## Author Contributions

**Conceptualization:** Nicole Bertges, Sachiyo Shearman, Satomi Imai, Jo Anne G. Balanay, Sinan Sousan.

**Data curation:** Nicole Bertges, Sachiyo Shearman, Satomi Imai, Jo Anne G. Balanay, Sinan Sousan.

**Formal analysis:** Nicole Bertges, Sachiyo Shearman, Satomi Imai, Jo Anne G. Balanay, Sinan Sousan.

**Funding acquisition:** Sachiyo Shearman, Satomi Imai, Jo Anne G. Balanay, Sinan Sousan.

**Investigation:** Nicole Bertges, Sachiyo Shearman, Satomi Imai, Jo Anne G. Balanay, Sinan Sousan.

**Methodology:** Nicole Bertges, Sachiyo Shearman, Satomi Imai, Jo Anne G. Balanay, Sinan Sousan.

**Project administration:** Sinan Sousan.

**Software:** Nicole Bertges.

**Supervision:** Sachiyo Shearman, Satomi Imai, Jo Anne G. Balanay, Sinan Sousan.

**Validation:** Nicole Bertges, Sachiyo Shearman, Satomi Imai, Jo Anne G. Balanay, Sinan Sousan.

**Visualization:** Nicole Bertges, Sinan Sousan.

**Writing – original draft:** Nicole Bertges, Sachiyo Shearman, Satomi Imai, Jo Anne G. Balanay, Sinan Sousan.

**Writing – review & editing:** Nicole Bertges, Sachiyo Shearman, Satomi Imai, Jo Anne G. Balanay, Sinan Sousan.

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
