## [Decision Letter · Decision Letter 0]

2 Jul 2024

PONE-D-24-17804Examination of Factors that Impact Mask or Respirator Purchase and Usage During the COVID-19 PandemicPLOS ONE

Dear Dr. Sousan,

Thank you for submitting your manuscript to PLOS ONE. After careful consideration, we feel that it has merit but does not fully meet PLOS ONE’s publication criteria as it currently stands. Therefore, we invite you to submit a revised version of the manuscript that addresses the points raised during the review process.

Please pay attention on following points also :Refine introductionMention sample size and elaborate methodology section for more clear understandingClearly define the basis on which participants were selected.

We look forward to receiving your revised manuscript.

Kind regards,

Sadia Ilyas, Ph.D.

Academic Editor

PLOS ONE

 [The National Institute of Environmental Health Sciences of the National Institutes of Health under Award Number P30ES025128.].  

3. In the online submission form, you indicated that [All data are availble upon request from the authors]. 

Reviewers' comments:

Reviewer's Responses to Questions

**Comments to the Author**

1. Is the manuscript technically sound, and do the data support the conclusions?

Reviewer #1: Yes

Reviewer #2: Yes

2. Has the statistical analysis been performed appropriately and rigorously? 

Reviewer #1: Yes

Reviewer #2: N/A

3. Have the authors made all data underlying the findings in their manuscript fully available?

Reviewer #1: Yes

Reviewer #2: Yes

4. Is the manuscript presented in an intelligible fashion and written in standard English?

Reviewer #1: Yes

Reviewer #2: Yes

5. Review Comments to the Author

Reviewer #1: Dear Authors,

I had the opportunity to review your manuscript titled 'Examination of Factors that Impact Mask or Respirator Purchase and Usage During

the COVID-19 Pandemic,' and I commend you on your thorough investigation and reporting. I appreciate the opportunity to provide feedback on your work."

However, I have provided some feedback on other aspects of the manuscript, particularly the methodology and discussion sections. Please find my comments and suggestions outlined below:

Methodology:

1. As you already mentioned the exclusion criteria in lines 111-112 “Sample selection excluded individuals below the age of 18 and non-English speakers” mentioning it again as the inclusion criteria is not needed in line 116 “were only 116 English speakers and adults older than 18”

2. Please rewrite the Table 1 title as Variables Analyzed instead of “Dependent Variables Analyzed”

3. In “Table 1: Dependent Variables Analyzed”what is the basis of categorizing the income of the respondent into two groups “Under $ 35,000 annually” and “Over $35,000 annually”

4. Please mention how the participants were recruited.

5. Please mention if the survey questionnaire was pretested or validated prior to the data collection.

6. Please clearly mention the study design at the start of the Methods section to help readers quickly understand the approach.

7. Kind describe shortly regarding the sample size was calculation. Although some information was mentioned in the Limitations sections, it would be best to include a concise description in the methodology as well.

Discussion:

1. In line 214 it is mentioned that “These results are interesting because females have been previously associated with a higher frequency of mask-wearing behavior” and the following paper was cited “McMahon E, Wada K, Dufresne A. Implementing fit testing for N95 filtering facepiece respirators: practical information from a large cohort of hospital workers. Am J Infect Control. 2008;36(4):298-300. doi:10.1016/j.ajic.2007.10.014”. However, this article does not contain any supporting findings. Please cite reference number 31 “Looi, K.H. Explicating gender disparity in wearing face masks during the COVID-19 pandemic. BMC Public Health 22, 2273 (2022). https://doi.org/10.1186/s12889-022-14630-7” which has relevant findings.

2. In line 224-226 it is mentioned that “Previous studies have associated headaches, itching, increased facial temperatures, and humidity with the 195 N95 and could be contributing to some choosing looser fitting masks over respirators” followed by in text citation of reference number 30 and 32. However, reference number 30 does not have any supporting findings. Please remove reference number 30 [“Christopher L, Rohr-Kirchgraber T, & Mark S.. The PPE pandemic: Sex-related discrepancies of N95 mask fit. European Medical Journal. https://www.emjreviews.com/microbiology-infectious-diseases/article/the-ppe 428 pandemic-sex-related-discrepancies-of-n95-mask-fit-j100121/. Published 2021 Dec. 22. Accessed 2023 Aug. 15”] and keep only 32 [“Scheid JL, Lupien SP, Ford GS, West SL. Commentary: Physiological and sychological Impact of Face Mask Usage during the COVID-19 Pandemic. Int J Environ Res Public Health. 2020;17(18):6655. Published 2020 Sep 12. doi:10.3390/ijerph17186655”] here, or if needed add other relevant references.

Your manuscript demonstrates considerable promise, and I am confident that addressing these points will enhance the robustness of your research.

Reviewer #2: The article is on impact of various motivational factors on individual choice of masks or respirators during the COVID-19 pandemic. The article is important to scientific community, however lacks many important details and analysis. It can be improved by heeding my comments, suggestion, and queries.

1) The 1st sentence of the introduction mentions the place from where the pandemic emerged. There is still lack of proof of source of COVID-19, and mentioning a certain city as source may not be appropriate. The source is considered to be from a non-human species which could be native to any country. Instead of direct reference, use words like 'may have', or 'most probably have emerged'

2) The requirement and selection of masks and respirators cannot be generalised to USA only. different nations have been using masks on daily basis even pre COVID-19. The current study is only valid for the 619 English speaking participants and their country.

3) Air flow resistance has less to do with masks then with respiratory disease. The term 'airflow resistance' should be removed. Stick only with 'breathability'.

4) The exclusion criteria is already mentioned in lines 111-112 “Sample selection excluded individuals below the age of 18 and non-English speakers” mentioning it again as the inclusion criteria is not needed in line 116 “were only 116 English speakers and adults older than 18”

5) There is no clarity on how the participants were recruited. please clarify this.

6) Please give explanation on the validation of data collection.

7) The methodology is not defined clearly, which creates confusion to go through the manuscript. also, mention the sample size to show the reliability of the study.

6. PLOS authors have the option to publish the peer review history of their article (what does this mean?). If published, this will include your full peer review and any attached files.

Reviewer #1: No

Reviewer #2: No

---

## [Author Response · Author response to Decision Letter 0]

19 Jul 2024

We have provided a point-by-point response document that addresses all the reviewer's comments.

---

## [Editor Report · Decision Letter 1]

23 Jul 2024

PONE-D-24-17804R1Examination of Factors that Impact Mask or Respirator Purchase and Usage During the COVID-19 PandemicPLOS ONE

Dear Dr. Sinan Sousan,

Thank you for submitting your manuscript to PLOS ONE. After careful consideration, we feel that it has merit but does not fully meet PLOS ONE’s publication criteria as it currently stands. Therefore, we invite you to submit a revised version of the manuscript that addresses the points raised during the review process.

1. Additionally**, **please provide explanation of data validation.2. Please compact abstract and conclusion part.

We look forward to receiving your revised manuscript.

Kind regards,

Sadia Ilyas, Ph.D.

Academic Editor

PLOS ONE
---

## [Author Response · Author response to Decision Letter 1]

29 Jul 2024

We have provided a point by point response document to address the comments

---

## [Editor Report · Decision Letter 2]

1 Aug 2024

Examination of Factors that Impact Mask or Respirator Purchase and Usage During the COVID-19 Pandemic

PONE-D-24-17804R2

Dear Dr. Author,

We’re pleased to inform you that your manuscript has been judged scientifically suitable for publication and will be formally accepted for publication once it meets all outstanding technical requirements.

Kind regards,

Sadia Ilyas, Ph.D.

Academic Editor

PLOS ONE
---

## [Editor Report · Acceptance letter]

7 Aug 2024

PONE-D-24-17804R2 

PLOS ONE

Dear Dr. Sousan, 

I'm pleased to inform you that your manuscript has been deemed suitable for publication in PLOS ONE. Congratulations! Your manuscript is now being handed over to our production team.

Kind regards, 

on behalf of

Prof. Sadia Ilyas 

Academic Editor

PLOS ONE